# Framing the potential of public frameshift peptides as immunotherapy targets in colon cancer

Ide T. Spaanderman[1,2], Fleur S. Peters[2], Aldo Jongejan[3], Egbert J. W. Redeker[4], Cornelis J. A. Punt[1,2,3,4,5], Adriaan D. Bins[1,2]*

**1** Dept. of Medical Oncology, Amsterdam University Medical Centers, University of Amsterdam, Cancer Center Amsterdam, Amsterdam, The Netherlands, **2** Dept. of Experimental Immunology, Amsterdam University Medical Centers, University of Amsterdam, Amsterdam Infection & Immunity, Amsterdam, The Netherlands, **3** Dept. of Bio-informatics, Amsterdam University Medical Centers, University of Amsterdam, Amsterdam, The Netherlands, **4** Dept. of Clinical Genetics, Amsterdam University Medical Centers, University of Amsterdam, Amsterdam, The Netherlands, **5** Dept. of Epidemiology, Julius Center for Health Sciences and Primary Care, University Medical Center Utrecht, Utrecht, The Netherlands

* a.d.bins@amsterdammumc.nl

**Data Availability Statement:** TCGA-data is available trough the TCGA initiative at https://portal.gdc.cancer.gov/. The proprietary Perl computer code used to calculate PTC position in coding

## Abstract

Approximately 15% of Colon Cancers are Microsatellite Instable (MSI). Frameshift Peptides (FPs) formed in MSI Colon Cancer are potential targets for immunotherapeutic strategies. Here we comprehensively characterize the mutational landscape of 71 MSI Colon Cancer patients from the cancer genome atlas (TCGA). We confirm that the mutations in MSI Colon Cancers are frequently frameshift deletions (23% in MSI; 1% in microsatellite stable), We find that these mutations cluster at specific locations in the genome which are mutated in up to 41% of the patients. We filter these for an adequate variant allele frequency, a sufficient mean mRNA level and the formation of a Super Neo Open Reading Frame (SNORF). Finally, we check the influence of Nonsense Mediated Decay (MMD) by comparing RNA and DNA sequencing results. Thereby we identify a set of 20 NMD-escaping Public FPs (PFPs) that cover over 90% of MSI Colon, 62.2% of MSI Endometrial and 58.8% of MSI Stomach cancer patients and 3 out of 4 Lynch patients in the TCGA-COAD. This underlines the potential for PFP directed immunotherapy, both in a therapeutic and a prophylactic setting in multiple types of MSI cancers.

## Introduction

Protein products derived from mutated regions of DNA are a source of neo-antigens that can drive immune activation against cancer and convey susceptibility to immunotherapy [1–4]. Specifically, frameshift insertions and deletions (INDELS) are able to drive the anti-tumor immune response [5,6]. INDELS in coding DNA result in frameshifted RNA containing neo open reading frames (NORFS) that, once translated, give rise to completely non-self, out-of-frame protein products. These out-of-frame proteins can be degraded into peptides and presented in the context of multiple MHC alleles. Recognition of these MHC-peptide complexes

microsatellites is available at https://github.com/
ITSpaanderman/PTC_predictor, including run
instructions and test samples. GATK pipeline
information is available at https://gatk.
broadinstitute.org/hc/en-us. RSEM computer code
is made available by its author at https://github.
com/deweylab/RSEM.

**Funding:** A.D. Bins received the Dutch Cancer
Society (KWF) Grant 21923. The funder had no role
in study design, data collection and analysis,
decision to publish, or preparation of the
manuscript.

**Competing interests:** The authors have declared
that no competing interests exist.

by CD4+ or CD8+ T-cells initializes an anti-tumor immune response and leads to an influx of T cells into the tumor microenvironment [1,7,8], Longer NORFS, consisting of at least 50 aminoacids, so called super NORFS (SNORFS), have a higher chance to be recognized as a neoantigens [9]. Importantly, NORFS are subject to degradation on a RNA level by the Nonsense Mediated Decay (NMD) pathway [10]. NMD can be a limiting factor in tumor immunity, which can be resolved by direct targeting of the NMD pathway [11,12]. In line with this, the presence of NMD escaping SNORFS has been shown to be a strong predictor of immunotherapy response and survival in multiple cancer types [9].

Compared to other mutation signatures, INDELS are overrepresented in the exome of microsatellite instable (MSI) tumors [5,13]. Microsatellites are regions in the DNA with a repeated nucleotide motif. These regions are prone to polymerase slippage, release and subsequently incorrect re-annealing during DNA synthesis. This leads to single stand insertions and deletions in the nascent DNA strand [14]. In healthy cells, the mismatch repair (MMR) system recognizes and repairs these damages. However, MSI tumors have a deficient MMR system (dMMR) due to germline and/or acquired somatic mutations in one or multiple MMR genes (*MLH1*, *MSH2*, *MSH6*, *PMS2*, *EPCAM*) or by hyper methylation of the *MSH1* promotor [15,16]. Approximately 15% of colon cancers are considered MSI [17]. These dMMR colon cancers have high numbers of mutations, specifically INDELS in microsatellites.

Here we comprehensively assess frameshift mutations in colon cancer samples sequenced within the genome atlas (TCGA-COAD) [18]. We focus on the TGCA-COAD series as this contains a large subset of MSI tumors. Using a novel in silico approach (**Fig 1**), we establish that colon cancer frameshift mutations cluster to specific loci in the colon cancer exome. Based on DNA and RNA sequencing data we investigate whether these mutations lead to SNORFS and subsequently whether these SNORFS are affected by NMD. Finally, we identify a set of frequently occurring frameshift peptides (FPs), that we name public frameshift peptides (PFPs). These PFPs are potential targets for immunotherapy strategies in an adjuvant setting, as they obviate the need for time-consuming personalization of vaccines. Moreover, PFPs can also be found in all four (Lynch) patients with a germline MMR mutation in the TCGA--COAD. This highlights the possibility for PFP specific vaccination in a prophylactic setting as well.

## Results

### Frameshift mutations cluster to specific genomic loci in MSI colon cancer

In line with previous reports in literature 71 out of 461 (15%) of TCGA-COAD patients are MSI [17]. The majority of these patient are sporadic MSI (n = 67, 94.4%), but a small number of patients have germline mutations in the mismatch repair pathway and are therefore considered Lynch patients (n = 4, 5.6%). As expected, in general MSI colon cancer patients have a higher mutational burden than microsatellite stable (MSS) colon cancer patients (MSI: median = 2974, n = 71; MSS: median = 218, n = 272; p < 0.0001) (**Fig 2A**) and a larger percentage of their mutations consists of deletions or insertions (MSI: SNP = 72%, INS = 5%, DEL = 23%'; MSS: SNP = 97%, INS 1%, DEL = 1%) (**Fig 2B**). Almost all of these INDELS are frameshift mutations (INS: Frameshift 99%; DEL: Frameshift 98%) (**Fig 2C**). Contrary to single nucleotide polymorphisms (SNPs), these INDELS are preferentially located in coding microsatellites (cMS). (Median cMS length on mutation locus INS = 7; DEL = 7; SNP = 1, p < 0.0001) (**Fig 2D**). This highlights the role of the dysfunctional MMR system in the mutation signature of MSI colon cancer. As expected, a subset of mutations cluster to specific loci in the MSI colon cancer exome, similar to the well documented BRAF SNP at chr7:140753336

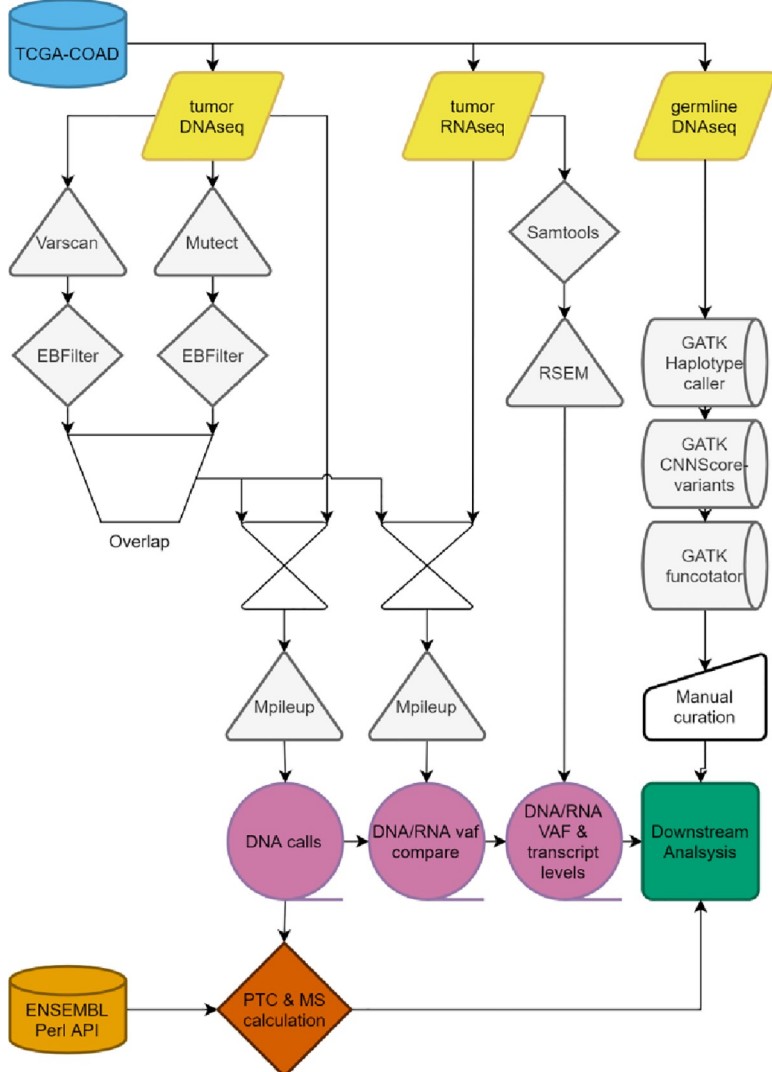

**Fig 1. Schematic overview of *in silico* pipeline.** TCGA-COAD: The cancer genome atlas colorectal adenocarcinoma database; Varscan: Somatic variant calling; Mutect: Somatic variant calling; EBFilter: Bayesian filtering for somatic variants; Mpileup: Location specific read calling; RSEM: Transcription quantification; VAF: Variant allele frequency; ENSEMBL Perl API: Programmable interface for Ensembl genome browser; GATK: Genome analysis toolkit; Haplotypecaller: Genetic variant calling; CNNScoreVariants: Germline variant filtering; Functotator: Clinvar based annotator of genetic variants; PTC: Premature termination codon; MS: Microsatellite.

[19], it seems probable that the most prevalent mutations also play a role in tumorigenesis, (**Fig 3A**). 13 of the 14 most prevalent mutations are frameshift mutations located in a cMS. Strikingly, the most frequently occurring frameshift mutation is present in 46% of the MSI patients. 123 frameshift mutations occur in more than 10% of the patients (**Fig 3B**). Notably 82.5% of the deletions that occur in more than 10% of patients are listed in the candidate cancer gene atlas and 42.5% are listed as potential drivers in colon cancer [20] (**S1 Table**). Interestingly, when comparing the tendency to cluster among DEL, INS and SNPs mutations, clustering (with the exception of the BRAF SNP) is only clearly present for DEL mutations (median percentage of patients with a mutation on exactly the same locus in the genome; DEL = 14.08%, INS = 4.23%, SNP = 4.23%, $p < 0.0001$) (**Fig 3C**).

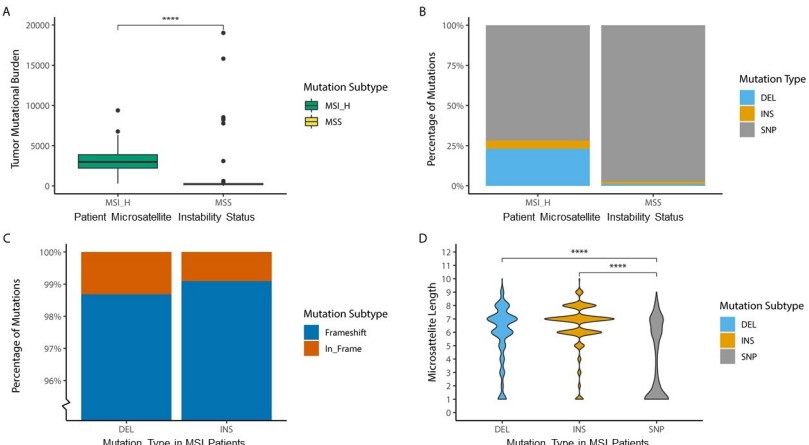

**Fig 2. Mutational profile for MSS and MSI in TCGA-COAD. A)** Difference in tumor mutational burden as number of mutations per patient between MSI (median = 2974, n = 71) and MSS patients (median = 218, n = 272, p < 0.0001). **B)** Mutation types as percentage of mutations between MSI (DEL = 23.15%, INS = 5.06%, SNP = 71.19%, n = 71) and MSS patients (DEL = 1.21%, INS = 1.34%, SNP = 97.4%, n = 218) **C)** Percentage of frameshift mutations for deletion (Frameshift = 98.69%, In_Frame = 1.31%, n = 53851) and insertions (Frameshift = 99.07%, In_Frame = 0.93%, n = 13172) in MSI patients (n = 71). **D)** Difference in microsatellite length in coding mononucleotide repeats on location of mutation in the genome between mutation subtypes, DEL (median = 7, n = 4960), INS (median = 7, n = 2384), SNP (median = 3, n = 1859, p < 0.0001) in MSI patients (n = 71). Microsatellite length of 3 mononucleotide repeats or less cannot be considered true microsatellites.

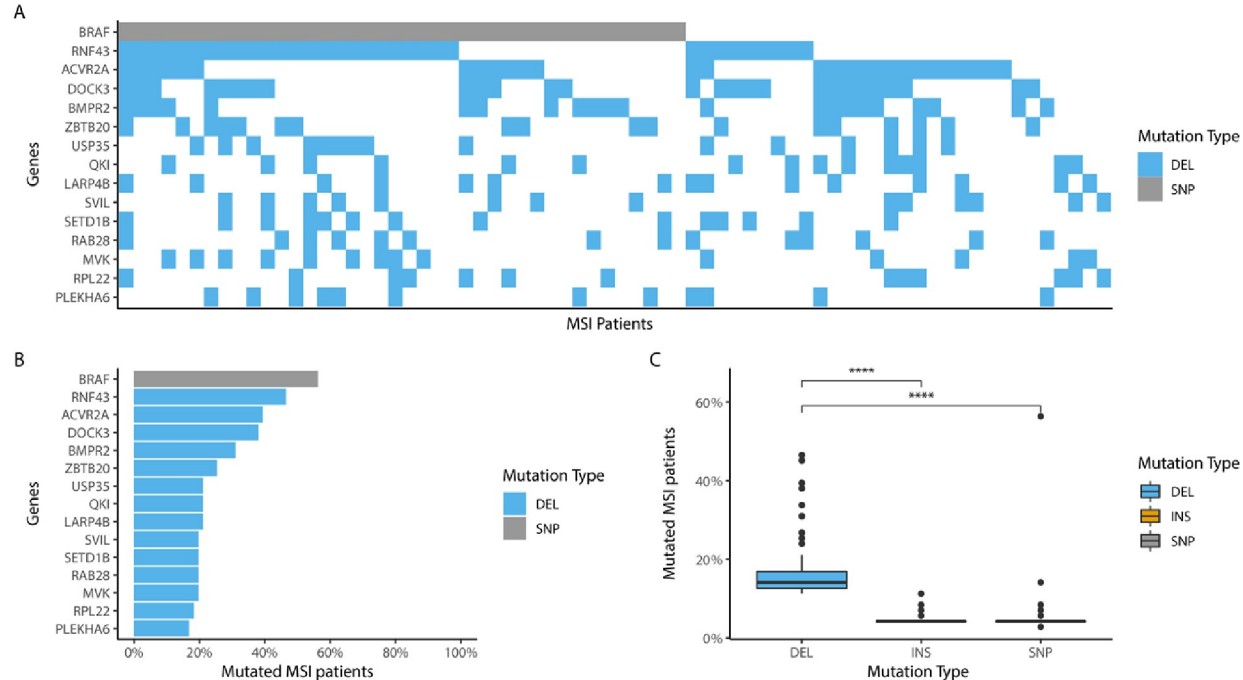

**Fig 3. Mutation clustering in MSI TCGA-COAD. A)** Waterfall graph depicting the 15 most frequently occurring mutation hotspots by Gene Identifier in MSI patients. Light grey mutations are SNPs. Light blue mutations are DELs. Other mutation types do not occur within the 15 most frequently occurring mutation hotspot. **B)** Percentage of most frequently occurring mutation loci. **C)** Difference in mutation frequency as percentage of patients with mutation on locus between mutation types for the top 100 most frequently occurring mutations, DEL (median = 14.08%, n = 100), INS (median = 4.23%, n = 100), SNP (median = 4.23%, n = 100, p < 0.0001) in MSI patients (n = 71).

## Nonsense mediated decay escape

As previously mentioned, NORFS derived from INDELS are potentially subject to NMD. NMD is triggered by the formation of a premature termination codon (PTC), that is prone to occur in out-of-frame NORFS [10]. Such a PTC can make the mutated transcript susceptible to degradation by NMD. One major rule that defines this is the location of the PTC. If such a PTC is located at least 50 base pairs before the last exon-exon junction, the mRNA is considered sensitive to degradation by NMD. If the PTC is located behind this point the transcript is considered NMD resistant [21]. A complete set of rules can be applied to predict whether a mutation leads to a NMD sensitive PTC [11]. In the MSI cancers of the TGCA-COAD this concept holds true, as the median Z-score for mutated transcripts, the difference between mutated mRNA transcript level and the mean transcript level, with a NMD sensitive PTC is significantly lower than that of a mutated transcripts with a NMD resistant PTC (median SENS = -0.16; RES = 0.14; p < 0.0001) (**Fig 4A**). Nonetheless the NMD effect is limited in an absolute sense and clearly heterogeneous as a large portion of potential sensitive mutations seem to escape NMD based on the Z-score of mRNA transcript levels. However, the fact that a single specific mutation can be mapped to multiple transcripts (isoforms), might limit the sensitivity for detection of transcript specific NMD signals.

To negate this problem, we checked the Variant Allele Frequency (VAF) for each mutated locus in the DNA and RNA. The VAF is the percentage of mutated reads in all reads on that specific locus. We calculated the percentage percentile difference between DNA and RNA by dividing the DNA VAF by the RNA VAF: the VAF-ratio. This VAF ratio is transcript independent and therefore a much more sensitive and accurate measurement of NMD (**S1 Fig**). On average, for NMD sensitive mutations the variant allele frequency (VAF) in RNA sequencing data is decreased by 42.1% compared to the DNA VAF (median VAF-ratio 57.9 n = 1324), which is significantly more than the NMD resistant mutations (median VAF-ratio 100%, n = 685; p < 0.0001) (**Fig 4B**). This clearly confirms that NMD is an important factor in the formation of neo-antigens from NORFS in MSI colon cancer. Nonetheless roughly 75% of NMD sensitive NORFS still escape NMD, with a VAF-ratio of 75% or more. This presents a clear opportunity for PFP formation.

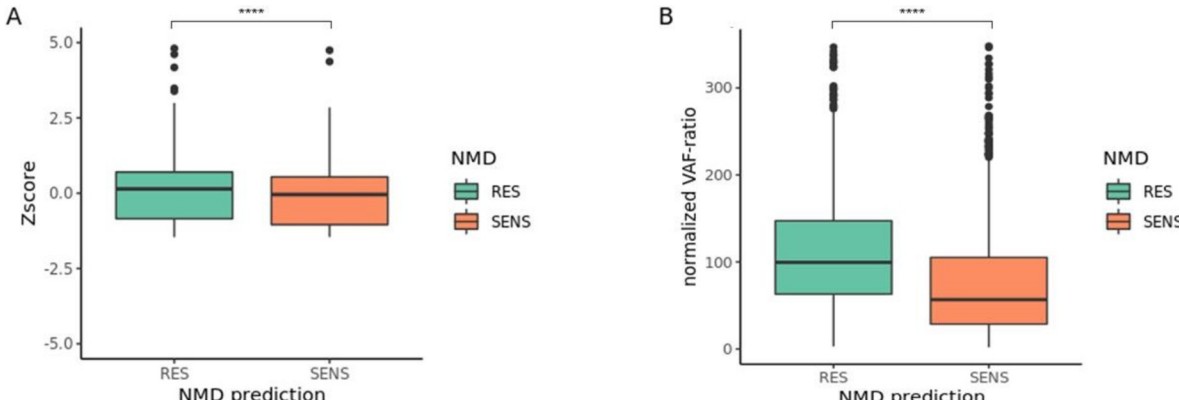

**Fig 4. Nonsense mediated decay escape in MSI TCGA-COAD.** Approximation of Nonsense Mediated Decay(NMD) on mutated transcripts either predicted as NMD sensitive (SENS) or NMD resistant (RES), based on the location of the premature termination codon (PTC) by NMDdetective **A)** NMD defined as a z-score of transcript levels for RES (median = 0.14, n = 973) and SENS (median = -0.16, n = 2224, p < 0.0001). **B)** NMD defined as normalized VAF-ratio: the allele frequency (VAF) as percentage RNA VAF of DNA VAF for RES (median = 100%, n = 685) and SENS (median = 57.9%, n = 1324, p < 0.0001).

## NMD escape by public frameshift peptides

Besides the potential NMD impact of PTCs, PTCs also can limit NORF length to a size that prohibits presentation of the full size of the frameshifted region in the context of MHC. Peptides that are presented in MHCI are generally at least 9 amino acids in length [22], but MHCII binding peptides are highly variable in length [23]. In the NORFS identified by us in MSI cancers of the TGCA-COAD, a PTC occurs on average after 14 amino acids (**S2 Fig**). Therefore, we selected potential PFP by discarding NORFS with a length below 10 amino acids. Hereby, we identified 90 SNORFS occurring in more than 5% of MSI patients. Relying on our own experience that a set of 20 PFPs is feasible to incorporate in a multivalent vaccine [23], we discarded the 70 least frequently occurring SNORFS (**Table 1**). The remaining 20 frequently occurring SNORFS have a mean NORF length of 28 amino acids, a mean VAF-ratio of 67.93% and a mean mRNA level of 2.30 (Log2 TPM) for the most abundantly available corresponding transcript (isoform). Most importantly, at least one of these 20 SNORFS is present in the exome of 93% of MSI samples in the TGCA-COAD. A limited set of 10 SNORFS, as has been used in a personalized RNA vaccination setting [24], would still cover 74.6% of these patients (**S3 Fig**). This highlights the potential for non-personalized immunotherapy that targets PFP in MSI colon cancer patients. Notably, 25% of these 20 SNORFS are listed as potential drivers in colon cancer [20].

## PFP set covers germline mismatch repair deficient tumors in TCGA-COAD

Lynch patients carry a heterozygous germline mutation in an MMR gene, resulting in a 50–80% life-time chance to develop colon cancer [25]. MSI colon cancer tumorigenesis in Lynch patients may be different than that of sporadic MSI colon cancer, leading to expression of different PFPs. At the same time, prophylactic immunotherapy targeting PFP may be beneficial for these patients. Therefore we evaluated whether the PFP set may potentially be applicable to patients with a germline MMR mutation. Using normal blood DNA sequencing data, we identified 4 MSI TCGA-COAD patients (5.6%) with a pathogenic germline dMMR mutation. This percentage is in line with earlier reports of germline dMMR mutations in literature [25]. In overlap with sporadic MSI colon cancers, three out of four of these germline mutated dMMR tumors encodes at least 1 PFP from the 20-valent PFP-set. Colon Cancer samples with a germline MMR mutation from two patients only have 1 mutation in one of the identified hotspots and 1 germline mutated MSI sample has 4 hotspot mutations (**Table 1**). This supports the potential of the PFP-set in a prophylactic vaccination setting for Lynch patients.

## PFP set overlaps with majority of MSI stomach and MSI endometrial tumors

As large subsets of endometrial (TCGA-UCEC) and stomach tumors (TCGA-STAD) are also microsatellite unstable, the defined PFP might also be applicable in these patients. Recently it has been reported that frequently occurring INDEL mutations between MSI patients overlap, independent of tissue origin, suggesting a role in tumor evolution [26]. There indeed is a significant overlap between INDELS in the 10 most frequently mutated INDELS in these MSI cancer types (15–25%) and to a lesser extend in the top 100 most frequently mutated INDELS (13–14%) (**Fig 5A**). Most importantly this overlap does occur in both mutations predicted as NMD sensitive and resistant (**Fig 5B**), which provides potentially overlapping SNORFS between MSI cancer types. Ultimately the 20 PFP set defined in MSI colon cancer covers 62.2% of MSI endometrial (n = 74) and 58.8% of MSI stomach (n = 34) cancer patients.

**Table 1. Public Frameshift Proteins (PFP).**

| POSITION (HG38) | GENE | MS LENGTH | MS BASE | TYPE | NMD | PTC DISTANCE (AMINO ACIDS) | DELTAVAF (MEAN) | TRANSCRIPT MRNA LEVEL (LOG2 TPM) | CUMULATIVE MSI PATIENTS COVERED (N %) | cUMULaTIVE lynch pATIENTS COVERED (N, %) |
|---|---|---|---|---|---|---|---|---|---|---|
| 17:58357799 | RNF43 | 7 | G | Frameshift_Del | SENS | 39 | 85.73% | 2.37 | **33 (46.5%)** | |
| 12:121804751 | SETD1B | 8 | C | Frameshift_Del | RES | 28 | 51.65% | 2.30 | **36 (50.7%)** | |
| 2:97083987 | FAHD2B | 7 | C | Frameshift_Del | RES | 23 | 57.86% | 1.25 | **39 (54.9%)** | |
| x:56565440 | UBQLN2 | 7 | C | Frameshift_Del | RES | 71 | 103.06% | 3.30 | **44 (62.0%)** | 1 (25%) |
| 9:134053406 | BRD3 | 8 | C | Frameshift_Del | RES | 84 | 111.84% | 3.09 | **45 (63.4%)** | |
| 4:94252758 | SMARCAD1 | 8 | A | Frameshift_Del | SENS | 24 | 77.61% | 1.90 | **47 (66.0%)** | |
| 20:32202148 | PLAGL2 | 7 | C | Frameshift_Del | RES | 32 | 53.33% | 2.64 | **47 (66.2%)** | |
| 10:88922388 | STAMBPL1 | 8 | A | Frameshift_Del | RES | 20 | 98.90% | 2.44 | **50 (70.4%)** | |
| x:130056035 | BCORL1 | 7 | C | Frameshift_Del | RES | 20 | 129.17% | 1.22 | **52 (73.2%)** | |
| 1:156745311 | HDGF | 8 | T | Frameshift_Del | SENS | 56 | 66.77% | 6.11 | **53 (74.6%)** | |
| 4:2696449 | FAM193A | 7 | A | Frameshift_Del | SENS | 23 | 57.85% | 2.53 | **54 (76.1%)** | 1 (25%) |
| 12:49040708 | KMT2D | 5 | C | Frameshift_Del | SENS | 28 | 96.06% | 2.29 | **58 (81.7%)** | 3 (75%) |
| 21:14966008 | NRIP1 | 8 | A | Frameshift_Del | RES | 22 | 66.73% | 2.22 | **58 (81.7%)** | 3 (75%) |
| 22:17818165 | MICAL3 | 7 | C | Frameshift_Del | SENS | 104 | 68.38% | 1.72 | **59 (83.1%)** | |
| 13:52474898 | CKAP2 | 8 | A | Frameshift_Del | RES | 14 | 66.05% | 3.24 | **63 (88.7%)** | |
| 14:23276100 | HOMEZ | 8 | T | Frameshift_Del | RES | 12 | 98.17% | 2.11 | **64 (90.1%)** | |
| 19:13772153 | MRI1 | 6 | C | Frameshift_Del | RES | 24 | 61.77% | 2.15 | **65 (91.5%)** | 3 (75%) |
| 16:14252442 | MRTFB | 7 | C | Frameshift_Del | SENS | 42 | 121.43% | 1.94 | **65 (91.5%)** | |
| 1:27550357 | AHDC1 | 7 | A | Frameshift_Del | RES | 54 | 52.91% | 1.90 | **66 (93.0%)** | |
| 2:43225372 | ZFP36L2 | 6 | G | Frameshift_Del | RES | 41 | 67.48% | 6.38 | **66 (93.0%)** | |

Properties of the 20 most frequently occurring mutations with SNORF of at least 10 Amino Acids and DeltaVAF of 50%. MS: Microsatellite. Bold: Cumulative numbers of MSI patient covered by mutation (n = 71). Germline patients n = 4. NMD: Predicted NDM status based on PTC location in transcript by NMDetective. DeltaVAF: Mean RNA-VAF percentage of DNA-VAF for all mutated patients. Expression: log2(TPM + 1) for most abundant corresponding transcript.

## Discussion

Based on DNA and RNA sequencing data from MSI colon cancer patients in the TGCA we identified a subset of frameshift mutations that cluster to distinct cMS in the genome, produce SNORFS that evade NMD, and have sufficient length for presentation in MHC context. Based on this, we established a PFP-set of 20 frameshift peptides that cover 93% of individual MSI patients. Such a set of PFPs has potential for non-personalized cancer-specific immunotherapy

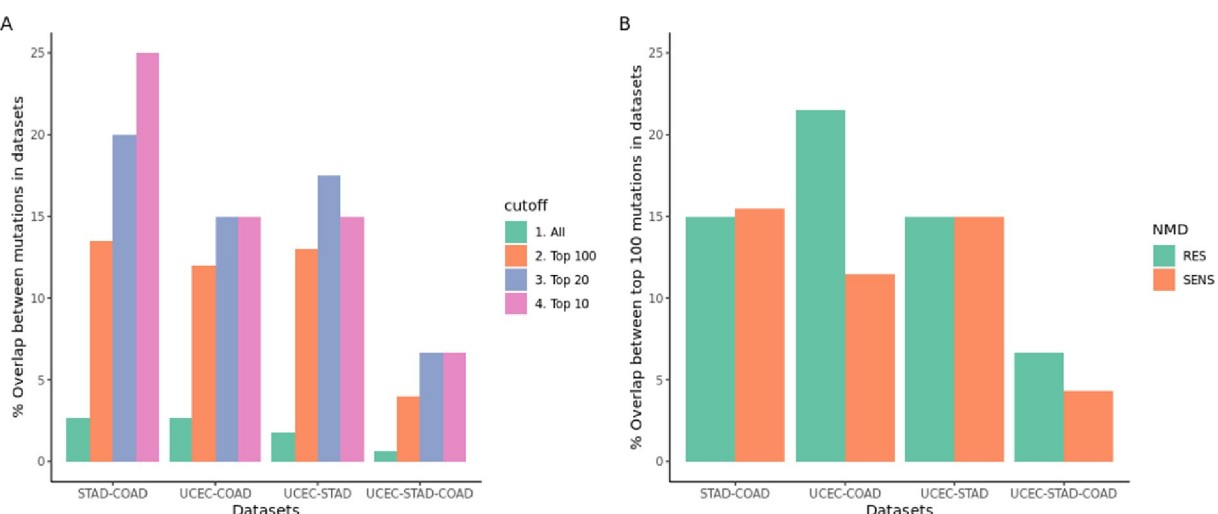

**Fig 5. INDEL mutation overlap in MSI tumors.** Overlap between INDEL mutations in MSI colon (COAD), MSI endometrial (UCEC) and MSI stomach (STAD) TCGA datasets **A)** differences in overlap for top 10, 20, 100 most frequent or all INDEL mutations **B)** overlap in top 100 most frequent mutated INDEL mutations split in NMD SENS and RES predicted mutations by NMDetective.

strategies directed against MSI colon cancer, especially in the adjuvant setting. In addition, this set of PFPs also covers three out of four Lynch patients in the TGCA-COAD, which supports potential application in a prophylactic vaccination setting for patients with Lynch syndrome. Clearly, as the sporadic MSI colon cancer mutation signature may differ from that of Lynch patients, target discovery for Lynch patients requires a larger patient cohort than the 4 patients identified in the TCGA-COAD.

We did not predict MHC binding for the selected PFPs. Currently MHC class I binding predictions tools only achieve approximately 40% sensitivity [27]. Although new methods have recently been published [28], especially the prediction of MHC class II binding remains very challenging, whilst MHC class II presented neo-antigens seem to drive anti-tumor immunity [28,29]. Furthermore, the above average length of the PFPs and the fact that they are non-self on any position makes adequate presentation within a naturally polymorphic MHC context more likely. Also, the concept of non-personalized immunotherapy would be compromised by MHC matching. In this context it is noteworthy that the loss of MHC-I on colon cancer cells could potentially limit the effectiveness of immunotherapy that target PFPs in the same way it can limit the effectiveness of checkpoint inhibition [30].

Finally, since this work is based on in silico analysis of sequencing data we could not determine the occurrence of PFPs on a protein level. Nonetheless, based on our extensive filtering for NMD signals in RNA sequencing data we deem it likely that a majority of the selected SNORFS will be translated and lead to presentable PFPs. Even SNORS with a DeltaVAF indicating a limited effect of NMD tend to harbor sufficient mutated mRNA. Also it has been shown that a first so-called pioneering round of translation, can already be adequate for neo-antigen formation [31]. In that regard our higher VAF threshold and strict NMD evasion filtering may have missed relevant PFPs. Furthermore, although we utilized novel NMD prediction algorithms [11], the availability of RNA sequencing data enables an unbiased determination of mutated RNA transcript levels, independent of PTC localization. In order to assess the PFP presentation on a protein level we are currently collecting patients samples in the ATAPEMBRO trial [32], which we aim to screen for specific PFP T-cell immunoreactivity by IFN-gamma Elispot.

In conclusion we identified a set of 20 DEL mutation hotspots in coding DNA microsatellites that give rise to adequately transcribed, NMD-escaping SNORFS and subsequently can produce PFPs. Likely, these SNORFs are involved in colon cancer tumorigenesis. The 20 most common PFPs cover 93% of MSI patients and 3 out of 4 patients that harbor a pathogenic germline MMR mutation in the TCGA-COAD. Furthermore these PFPs cover 62.2% of endometrial and 58.8% of stomach MSI patients. Therefore, these PFPs are potential targets for immunotherapy strategies, not only in in colon, but also in other MSI cancers.

## Methods

### Data acquisition, storage and computational analysis

All DNA and RNA sequencing data and corresponding patient characteristics, microsatellite panel and *MLH1* promotor methylation data was collected from the Cancer Genome Atlas Colon Adenocarcinoma(TCGA-COAD). Uterine Corpus Endometrial Carcinoma (TCGA-UCEC) and Stomach Adenocarcinoma (TCGA-STAD) databases [18]. Access to private data was granted by the National Cancer Institute Data Access Committee (NCIDAC) from the U.S. National Institute of Health trough the database of Genotypes and Phenotypes (dbGaP) portal. According to dbGaP guidelines private data was stored on the pre-approved internal Amsterdam UMC data-infrastructure. Computations on large public TCGA datasets were performed on the SurfSara high performance computing cloud (grant provided by the Dutch government).

### Somatic mutation calling from DNA sequencing files

Somatic mutations were called on aligned DNA reads by Mutect [33] and Varscan [34] separately with default quality control parameters. Those calls were independently filtered by empirical Bayesian mutation filtering using EBFilter [35] to reduce the false positive rate. Finally, the overlapping filtered mutations between these two methods for each individual patient were selected as true positive somatic mutations. VAF was calculated as percentage of total DNA reads that contains the mutation. A VAF cutoff of 20% was used.

### Microsatellite status and tumor mutational burden

Microsatellite status was retrieved directly from the TCGA-COAD dataset and computationally verified by MSIpred [36]. Tumor mutational burden (TMB) was calculated by dividing the total amount of exonic mutations by the length of the captured exome sequence for the particular sequencing experiment. A limited number of high TMB outliers in the MSS group can be identified (n = 5, 1.28%). These patients all have either one or two mutations in the POLE1 or POLE2 gene, well known for causing tumor hypermutation [37,38].

### Nonsense mediated decay escape by RNA and DNA mutated reads comparison (VAF-ratio)

Counts for mutated reads were collected by performing Mpileup [39] on matched RNA and DNA tumor aligned sequencing reads for each of the specific genomic position of the somatic mutations selected prior. Mutations with less than 10 mutated reads in the DNA were discarded, although this number was very limited due to the stringent selection of mutations. No lower limit was placed on the RNA read counts since this would hinder the detection of complete nonsense mediated decay (NMD). NMD is measured by the difference in percentage between the fraction of mutated reads in the DNA compared to the RNA (or $\Delta$VAF), were the

fraction of mutated reads in the DNA is set as 100%.

$$VAFratio \ (locus \ x \ for \ patient \ y) = \frac{Mutated \ Reads \ RNA \ x(y) \Big/ Total \ Reads \ RNA \ x(y)}{Mutated \ Reads \ DNA \ x(y) \Big/ Total \ Reads \ DNA \ x(y)} \times 100$$

transcripts that have a VAF-ratio of 50% or higher a considered NMD-escaping. For comparison between sensitive and resistant transcripts, all VAF scores are normalized against the median of all resistant transcripts

$$Example \ NMD \ resistant: \ \frac{^{150}/_{200}}{^{350}/_{400}} \times 100 = \frac{0.75}{0.875} \times 100 = 0.85 \times 100 = 85\%$$

$$Example \ NMD \ sensitive: \ \frac{^{50}/_{200}}{^{350}/_{400}} \times 100 = \frac{0.25}{0.875} \times 100 = 0.28 \times 100 = 28\%$$

## Transcript mRNA levels and Zscore

Accurate transcript quantification from RNA sequencing data is calculated by RSEM [40] and mapped to Ensembl [41] genome browser transcripts from reference genome GRCh38.p13. Absolute mRNA transcript levels per million (TPM) are log_2 transformed. The z-score is calculated for each individual mutated transcript. The Z-score of the mutated transcript in a specific patient is calculated by subtracting the mean mRNA level from the same (wild-type) transcript in all (MSI) patients from the mRNA level of the mutated transcript in the patient and subsequently dividing it by the standard deviation of this same transcript in all (MSI) patients.

$$Zscore \ (mutated \ transcript \ x \ for \ patient \ y) = \frac{mRNA \ x(y) - \mu \ mRNA \ x}{\sigma \ mRNA \ x}$$

Due to the internal control, the same transcript in the same type of patients, copy number alteration and gene expression variations do not influence this score. Nonetheless, because of the above factors, Z-scores can only be reliably be interpreted if calculated within a population with a single molecular cancer subtype, such as MSI Colon cancer or MSS Colon cancer. Transcript with a mean mRNA level less or equal than 1.0 are removed from the analysis, since these are considered not expressed in colon cancer tissue.

## Microsatellite and premature termination codon calculation

A proprietary Perl script run with GNU parallel utilizes the Ensembl Perl API to collect transcript sequences based on the position, allele, type and length of inputted true positive somatic mutations. Using this sequence, it calculates the number of mono-, di- or multi-nucleotide repeats on the mutation position. Subsequently it constructs the new reading frame after mutation and checks the RNA sequence until the occurrence of a premature termination codon. Finally, it records the position of this PTC in the transcript and relative to the mutational site.

## Germline mutation calling from normal tissue DNA sequencing files

Aligned DNA read files were spliced to incorporate only genomic regions of the mismatch repair genes *MLH1*, *MSH2*, *MSH6*, *PMS2* and *EPCAM*. Germline mutation calling on spliced aligned reads was performed using the GATK germline short variant discovery pipeline and using the GATK [42] best practices. The workflow subsequently utilizes Haplotypecaller,

CNNScoreVariants and FilterVariantTrances. Mutation calls with QD < 10 and read count < 10 were discarded. Mutational calls were functionally annotated employing Funcotator [42]. Variants of unknown significance were manually scored by a laboratory specialist clinical geneticist on a 1–5 scale according to the American College of Medical Genetics and Genomics (ACMG) guidelines [43], based on mutation characteristics, amino-acid change and taking patient clinical data, somatic mutations and MLH1 promotor methylation data into account.

### Downstream analysis

Statistical analysis were performed using R 3.4.4. in Rstudio 1.2.1335 (Rstudio, INC) with the tidyverse package enabled. Difference between means are calculated by a Mann Whitney U test for comparison between 2 groups and a Kruskall Wallis test for comparison between more than 2 groups. Graphs were generated using ggplot2.

## Supporting information

**S1 Fig. Difference in NMD measurement.** Density map showing difference between Z-score and VAF-ratio approach for the measurement of Nonsense Mediated Decay (NMD) in NMD RES and SENS mutations. Separation of NMD RES and SENS is clearer with VAF-ratio.
(DOCX)

**S2 Fig. Premature Termination Codon (PTC) distance.** PTC distance for mutation in MSI_H. Median is shown as dotted line (median = 14 Amino Acids). Dashed line shows 20 Amino Acids.
(DOCX)

**S3 Fig. Patient coverage for top 30 frameshift mutations.** Percentage of patients covered by n most frequently occurring frameshift mutations with adequate expression, a SNORF and NMD-escape.
(DOCX)

**S1 Table. Candidate cancer genes.** List with genes described in the candidate cancer gene database that are part of the top 100 most occurring mutations in MSI colon cancer. Rank: Relative rank assigned to CIS in study.
(DOCX)

## Acknowledgments

We would like to thank Prof. dr. L. Vermeulen for proofreading the manuscript.

## Code availability

The proprietary Perl computer code used to calculate PTC position in coding microsatellites is available at https://github.com/ITSpaanderman/PTC_predictor, including run instructions and test samples. GATK pipeline information is available at https://gatk.broadinstitute.org/hc/en-us. RSEM computer code is made available by its author at https://github.com/deweylab/RSEM.

## Author Contributions

**Conceptualization:** Ide T. Spaanderman, Fleur S. Peters, Cornelis J. A. Punt, Adriaan D. Bins.

**Data curation:** Ide T. Spaanderman, Aldo Jongejan.

**Formal analysis:** Ide T. Spaanderman, Adriaan D. Bins.

**Funding acquisition:** Adriaan D. Bins.

**Investigation:** Ide T. Spaanderman.

**Methodology:** Ide T. Spaanderman, Aldo Jongejan, Adriaan D. Bins.

**Resources:** Egbert J. W. Redeker.

**Software:** Ide T. Spaanderman, Aldo Jongejan.

**Supervision:** Cornelis J. A. Punt, Adriaan D. Bins.

**Validation:** Ide T. Spaanderman, Egbert J. W. Redeker.

**Writing – original draft:** Ide T. Spaanderman.

**Writing – review & editing:** Ide T. Spaanderman, Fleur S. Peters, Aldo Jongejan, Egbert J. W. Redeker, Cornelis J. A. Punt, Adriaan D. Bins.

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
