## [Decision Letter · Decision Letter 0]

28 Jul 2020

PONE-D-20-19308

Framing the potential of public frameshift peptides as immunotherapy targets in colon cancer

PLOS ONE

Dear Dr. Bins,

Thank you for submitting your manuscript to PLOS ONE. After careful consideration, we feel that it has merit but does not fully meet PLOS ONE’s publication criteria as it currently stands. Therefore, we invite you to submit a revised version of the manuscript that addresses the points raised during the review process.

We look forward to receiving your revised manuscript.

Kind regards,

Jianjiong Gao

Academic Editor

PLOS ONE

Journal Requirements:

2. Please describe in more detail what categories of private data you collected and how another researcher would obtain the data.

Reviewers' comments:

Reviewer's Responses to Questions

**Comments to the Author**

1. Is the manuscript technically sound, and do the data support the conclusions?

Reviewer #1: Yes

Reviewer #2: Yes

2. Has the statistical analysis been performed appropriately and rigorously? 

Reviewer #1: Yes

Reviewer #2: Yes

3. Have the authors made all data underlying the findings in their manuscript fully available?

Reviewer #1: Yes

Reviewer #2: Yes

4. Is the manuscript presented in an intelligible fashion and written in standard English?

Reviewer #1: Yes

Reviewer #2: Yes

5. Review Comments to the Author

Reviewer #1: This manuscript by Bins et al. highlights the potential of using SNORFS as a non-personalized immunotherapy in MSI colon cancer patients, as 10 SNORFS would still cover >=75% of the patients. It uses a large TCGA database with a reasonable number of MSI tumors (76 out of 461) colon cancers, although I found it a bit unusual that they did not consider other MSI cancer types in TCGA (point #1 below). They applied stringent and well-reasoned filters to the data (see however concern #2). However it seems that not all known NMD rules are represented therefore the NMD efficiency is not estimated as accurately as it could be, and this might be improved to bring the method to the state-of-the-art (see #3). One weakness of the study -- which I do not necessarily expect them to address, but they should at least discuss -- is that there was no replication/validation in an independent dataset. Overall their conclusions seem well supported with data and well presented. The very high recurrence of some deletions in MSI colon cancer is indeed potentially a source of near-universal neoantigens that is worth reporting.

Major remarks:

1. In addition to colon cancers, also the uterine and stomach cancers in TCGA are commonly MSI. The same analysis could be used to catalogue public frameshift peptides in those cancer types as well, with minimal alterations to their pipeline. It is a missed opportunity that this was not done, particularly so because it is of interest to compare the neopeptide repertoire across tissues (does it vary more than across individuals?)

2. Line 73: "This transcript independent method is a much more sensitive and accurate measurement of NMD" (I suppose they mean than their Z-score method: please make that clear and also explain Z-score better in Methods). Their Z-score does not appear control for copy number altered segments, nor for the confounding of gene expression by molecular subtype, even though these steps were perfomed in previous work (eg. PMID: 27618451 that they cite). Either refine the Z-score calculation, or omit it completely from the paper stating you rely on the VAF-ratio method only.

2b. Related to the above: According to VAF-ratio definition, higher values indicate that PTC-containing RNA is more degraded (by NMD). How do they explain that VAF ratio for NMD-sensitive transcripts is smaller (34.3%) than NMD-resistant transcripts (59.9%)? This may need clarifying. It was also not clear to me how NMD-resistant transcripts are downregulated by 40%, however the difference between sensitive and resistant transcripts downregulation is quite small (65-40%). I am a bit worried that they might not be classifying NMD-resistant and sensitive transcripts properly (perhaps because of the #3 below).

3. As I understood, they only consider one NMD rule to assess if a PTC will or not trigger NMD (line 61): last exon + 50nt rule, which accounts for only a part of the known of the NMD-escaping variance (PMID: 27618451). It is a bit puzzling why they ignored the other rules because they have referenced this paper. That the other NMD-escape rules might have an effect is consistent with what they write (line 66): "Nonetheless the NMD effect is limited in an absolute sense and clearly heterogeneous as a large portion of potential sensitive mutations seem to escape NMD based on the Z-score of mRNA transcript levels.". One way to address this is to use predictions from previous NMD prediction methods (eg PMID: 31659324), which also consider these other rules.

Minor.

- Line 139: "Furthermore the availability of RNA sequencing data also negated the need for novel NMD prediction algorithms." Please clarify.

- Table 1: CUMALITIVE MSI PATIENTS COVERED / CUMALITIVE LYNCH PATIENTS COVERED (N, %) > CUMULATIVE

- Line 12: "presence of NMD escaping SNORFS has been shown to be a strong predictor of immunotherapy response and survival in multiple cancer types" Consider citing PMID: 20463739 and/or PMID: 31659324

- Line 13: Better explain difference between NORFS and SNORFS. Which is the nt/aa length boundary?

- Line 39: Statistics reported for Lynch and sporadic cases together? Clarify.

- Line 50: "46% of the patients" Did they mean "46% of the MSI patients?"

- Line 55: Appears a bit contradictory with sentence at line 46-47. Basically, in line 46 one gets the idea that all 3 types of mutation cluster to specific loci, but then it is said that INS and SNP hardly cluster. Consider rephrasing.

- Line 83: When they describe peptide size-limitation for loading into the MHC complex, they should mention some references. They claim that 10 aas is not enough, but 14 might be (as I understood the text), there should be a source cited for this.

- Line 87: "Assuming that a set of 20 PFPs is feasible to incorporate in a multivalent vaccine". What is this assumption based on? There is no reference.

- Line 92: SNOFRS > SNORFS

- Line 119: Cleary > Clearly

- Line 136: Alsco > Also,

- Line 140: FPF > PFP

- Line 162: Clarify that this is about DNA and not RNA (it is stated in the title but not in the text)

- Line 178: 2log > log_2

- Line 191: typo "incorporated"

Reviewer #2: Spaanderman et al present an analysis of public frameshifted peptides, in microsatellite instable colorectal cancer. The work is timely in nature, given the broad scientific and clinical interest in novel anti-cancer immunotherapeutics. Spaanderman and colleagues take a novel approach to this challenge, focusing on public neoantigens, recurrently found in a high proportion of patients. This avoids the costs and excessive effort of personalized vaccines, which are private to each patient. In this report the authors provide a concise but highly robust analysis of the most promising frameshift peptide targets in CRC. They show a cocktail vaccine of ~20 peptides could cover up to ~90% of MSI CRC patients, raising the prospect of off-the-shelf therapeutics or even prophylactic utility in Lynch syndrome cases. A good level of technical validation is presented in the paper, with targets assessed for mutation frequency in both sporadic & hereditary CRC, as well as mRNA expression levels, neo open reading frame length and susceptibility to NMD degradation. Minor corrections are suggested below, however this manuscript is of broad interested and should be supported for timely publication.

Minor comments:

1. The introduction lacks important detail on the immunogenicity data supporting MSI peptides as promising therapeutic targets, e.g. capability to elicit T cell response, e.g. https://pubmed.ncbi.nlm.nih.gov/23561850/. In addition, clinical data from MSI checkpoint inhibitor trials should be cited, where immunogenicity was demonstrated to be driven strongly by frameshift events, e.g. https://science.sciencemag.org/content/357/6349/409. Lastly, recent pre-clinical data from MSI models, which again show enriched immunogenicity (immune editing) directed towards indels should be referenced, e.g. https://science.sciencemag.org/content/364/6439/485

2. In figure 3b could the frequency of the top 15 mutations also be reported for other major MSI histologies (e.g. endometrium (UCEC), stomach (STAD))? MSI cases from these histologies are reported in sup data 1 of: https://www.nature.com/articles/ncomms15180#Sec22. If the % of MSI STAD and UCEC cases covered can be added this will help broaden the relevance of this work to multiple cancer types.

3. Could the authors comment on the extreme outliers in the MSS group in Fig1 (TMB ~15000-20000) – are these potentially misclassified as MSS by TCGA? A comment to explain this should be added to the methods.

4. Figure 3C is difficult to interpret – from Line55 – how is “similar specific mutation” defined? Within a certain +/- bp distance? This definition should be given in the main text for clarity purposes.

5. Regarding statistical tests, given TMB and other measures in the manuscript are likely not normally distributed, they should consider replacing t-tests with non-parametric tests (e.g. Mann Whitney U test). Similarly Kruskal-wallis test should be considered for Fig.3C.

6. Line24 typo – should be “assess”

7. Panel Fig. 2C – percentages on the y-axis are presented to two decimal places, which is a bit unnecessary, please reduce to 1 or no decimal places.

8. Figure 3 – gene ACVR2A appears to be offset versus the others, can this be corrected.

9. Line42-45 – This sentence is not clear and could be split into two, and also made the numerical data on line45 (2d) refers to microsatellite length.

6. PLOS authors have the option to publish the peer review history of their article (what does this mean?). If published, this will include your full peer review and any attached files.

Reviewer #1: No

Reviewer #2: No

---

## [Author Response · Author response to Decision Letter 0]

24 Jan 2021

Response to reviewer questions

Major remarks

Where applicable questions have been bundled. Reviewer #1 is I and #2 II

1. Analysis in STAD and UCEC MSI

I. In addition to colon cancers, also the uterine and stomach cancers in TCGA are commonly MSI. The same analysis could be used to catalogue public frameshift peptides in those cancer types as well, with minimal alterations to their pipeline. It is a missed opportunity that this was not done, particularly so because it is of interest to compare the neopeptide repertoire across tissues (does it vary more than across individuals?)

II. In figure 3b could the frequency of the top 15 mutations also be reported for other major MSI histologies (e.g. endometrium (UCEC), stomach (STAD))? MSI cases from these histologies are reported in sup data 1 of: https://www.nature.com/articles/ncomms15180#Sec22. If the % of MSI STAD and UCEC cases covered can be added this will help broaden the relevance of this work to multiple cancer types.

- This is a very important suggestion from both reviewers and therefore we would sincerely like to thank both reviewers for this question. It has led to an important and insightfull new analysis. In order to accommodate this analysis we added a new subsectionat the end of the results section and created a new figure (F5). This analysis shows that there is definite overlap between INDELS MSI cancer types, but most important the PFP subset also covers 62.2% of endometrial MSI and 58.8% of stomach MSI patients (from TCGA-COAD and TCGA-STAD). These new findings implicate that frameshift peptide vaccination could be a potential treatment in all MSI cancer types.

2. Clarification of Zscore

I. Line 73: "This transcript independent method is a much more sensitive and accurate measurement of NMD" (I suppose they mean than their Z-score method: please make that clear and also explain Z-score better in Methods). Their Z-score does not appear control for copy number altered segments, nor for the confounding of gene expression by molecular subtype, even though these steps were perfomed in previous work (eg. PMID: 27618451 that they cite). Either refine the Z-score calculation, or omit it completely from the paper stating you rely on the VAF-ratio method only.

Related to the above: According to VAF-ratio definition, higher values indicate that PTC-containing RNA is more degraded (by NMD). How do they explain that VAF ratio for NMD-sensitive transcripts is smaller (34.3%) than NMD-resistant transcripts (59.9%)? This may need clarifying. It was also not clear to me how NMD-resistant transcripts are downregulated by 40%, however the difference between sensitive and resistant transcripts downregulation is quite small (65-40%). I am a bit worried that they might not be classifying NMD-resistant and sensitive transcripts properly (perhaps because of the #3 below).

- We thank reviewer #1 for pointing out that our explanation of the Zscore and VAF-ratio concepts are insufficient in the manuscript. The transcript independent method is actually the VAF-ratio, where a lower VAF ratio indicates NMD. We have edited both the main text, the legend of figure 3 and the methods section to better reflect the above. As stated below the NMD classification has been replaced by NMDetective algoritm by Lindenboom et al. (PMID: 31659324) Considering the downregulation of NMD resistant mutation this is also a good questions, and we are thankfull that the reviewer made us realize that the explanation for this fact does not become very clear. Most likely this downregulation is due to the effect of the VAF-ratio method, that relies on the difference between the DNA and RNA VAF. The fact that DNA contains two alleles, whereas just one is copied to the RNA it becomes expected that even in the situation of NMD resistant genes the mean VAF-ratio is approximately 50%. We have added a clarification of this in the methods section.

- As the Zscore is calculated by determining the mRNA transcript level of an individual mutated transcript in an individual Patient which is then related to the mRNA levels in the same transcript in the rest of the patient group we feel that there is an adequate internal control that negates the influence of copy number alteration and gene expression variations.

3. NMD SENS or RES classification

I. As I understood, they only consider one NMD rule to assess if a PTC will or not trigger NMD (line 61): last exon + 50nt rule, which accounts for only a part of the known of the NMD-escaping variance (PMID: 27618451). It is a bit puzzling why they ignored the other rules because they have referenced this paper. That the other NMD-escape rules might have an effect is consistent with what they write (line 66): "Nonetheless the NMD effect is limited in an absolute sense and clearly heterogeneous as a large portion of potential sensitive mutations seem to escape NMD based on the Z-score of mRNA transcript levels.". One way to address this is to use predictions from previous NMD prediction methods (eg PMID: 31659324), which also consider these other rules.

- Indeed this is a very valid point. The NMD prediction has been reclassified in the manuscript using the NMD detective algorthm by lindeboom et al. (PMID: 31659324). This has not altered the conclusions in this manuscript. Even after applying these complete NMD rules, the NMD effect remains limited and heterogeneous on the RNA level even in the NMD sensitive transcripts.

Minor remarks

Reviewer #1

1. Line 139: "Furthermore the availability of RNA sequencing data also negated the need for novel NMD prediction algorithms." Please clarify.

- As suggested we implemented the relatively novel NMD algorithm by Lindeboom et al. in order to determine more precisely if a Frameshift mutation should be considered NMD resistant or Sensitive. Nonetheless the availability of RNA data for these same patients makes it possible to determine mRNA transcript levels from each mutated transcripts and the RNA variant allele frequency for each mutated transcript. These values are a direct and thereby more accurate representation of the degradation and remaining quantity of mutated RNA. This in the end provides the best insight in potential for neo-epitope formation and does not necessarily rely on NMD prediction algorithms.

2. Table 1: CUMALITIVE MSI PATIENTS COVERED / CUMALITIVE LYNCH PATIENTS COVERED (N, %) > CUMULATIVE

- Corrected

3. Line 12: "presence of NMD escaping SNORFS has been shown to be a strong predictor of immunotherapy response and survival in multiple cancer types" Consider citing PMID: 20463739 and/or PMID: 31659324

- Thank you for suppling the additional references. An additional line briefly stating this has been added to the introduction. Both references are included

4. Line 13: Better explain difference between NORFS and SNORFS. Which is the nt/aa length boundary?

- Different thresholds for SNORFS have been defined. The paper cited defines SNORFS as NORFS with a minimal length of 50 amino acids. Regarding the SNORFS defined in our manuscript we choose a cutoff of at least 10 amino acids (as stated in Results). This lower cutoff is based on the minimal out of frame length that is required for full out of frame presentation in the context of MHC. 

5. Line 39: Statistics reported for Lynch and sporadic cases together? Clarify.

- The statistics represent all MSI patients, including Lynch patients. This has been clarified in the manuscript.

6. Line 50: "46% of the patients" Did they mean "46% of the MSI patients?"

- This is correct and corrected in the manuscript.

7. Line 55: Appears a bit contradictory with sentence at line 46-47. Basically, in line 46 one gets the idea that all 3 types of mutation cluster to specific loci, but then it is said that INS and SNP hardly cluster. Consider rephrasing.

- Corrected. With exception for the well documented BRAF SNP only frameshift deletions tent to cluster.

8. Line 83: When they describe peptide size-limitation for loading into the MHC complex, they should mention some references. They claim that 10 aas is not enough, but 14 might be (as I understood the text), there should be a source cited for this.

- More context is added in section NMD escape by public frameshift citing two articles for both HMCI as MCHII binding peptide length.

9. Line 87: "Assuming that a set of 20 PFPs is feasible to incorporate in a multivalent vaccine". What is this assumption based on? There is no reference.

- This is based on our own experience with DNA vaccination techniques. Refrence to book chapter on DNA vaccination has been added

10. Line 92: SNOFRS > SNORFS

- Corrected

11. Line 119: Cleary > Clearly

- Corrected

12. Line 136: Alsco > Also,

- Corrected

13. Line 140: FPF > PFP

- Corrected

14. Line 162: Clarify that this is about DNA and not RNA (it is stated in the title but not in the text)

- Corrected

15. Line 178: 2log > log_2

- Corrected

16. Line 191: typo "incorporated"

- Corrected

Reviewer #2

1. The introduction lacks important detail on the immunogenicity data supporting MSI peptides as promising therapeutic targets, e.g. capability to elicit T cell response, e.g. https://pubmed.ncbi.nlm.nih.gov/23561850/. In addition, clinical data from MSI checkpoint inhibitor trials should be cited, where immunogenicity was demonstrated to be driven strongly by frameshift events, e.g. https://science.sciencemag.org/content/357/6349/409. Lastly, recent pre-clinical data from MSI models, which again show enriched immunogenicity (immune editing) directed towards indels should be referenced, e.g. https://science.sciencemag.org/content/364/6439/485

- Thank you for your input on this very relevant prior work. Some of the work was already referenced in the manuscript, non-referenced work has been implemented and referenced.https://pubmed.ncbi.nlm.nih.gov/23561850/ is already referenced in the introduction (ref 2)

- https://science.sciencemag.org/content/357/6349/409 is already referenced in the introduction (ref 4)

- https://science.sciencemag.org/content/364/6439/485 has been added as a reference, confirming the importance of mismatch-repair in anti-PD1 response.

2. Could the authors comment on the extreme outliers in the MSS group in Fig1 (TMB ~15000-20000) – are these potentially misclassified as MSS by TCGA? A comment to explain this should be added to the methods.

- This is a very interesting question, and we believe that indeed it would be something that might be unclear to many readers. These outliers most likely represent a specific group of MSS patients with a mutation in polη, that has been described in literature to lead to high TMB, but no increased indel formation. A new section regarding MSI/MSS, TMB and polη has been added to the Methods.

3. Figure 3C is difficult to interpret – from Line55 – how is “similar specific mutation” defined? Within a certain +/- bp distance? This definition should be given in the main text for clarity purposes.

- Thank you for pointing out that this is not clear. Itrefers to mutations that occur on the exact same location in the genome andthis clarification has been added to the manuscript.

4. Regarding statistical tests, given TMB and other measures in the manuscript are likely not normally distributed, they should consider replacing t-tests with non-parametric tests (e.g. Mann Whitney U test). Similarly Kruskal-wallis test should be considered for Fig.3C.

- Thank you for this valid point. We took your suggestion and recalculated the statistics if needed using a Mann-Whitney U test when comparing two groups and a Kruskall Wallis test when comparing multiple groups. This did not alter the conclusions in our manuscript

5. Line24 typo – should be “assess”

- corrected

6. Panel Fig. 2C – percentages on the y-axis are presented to two decimal places, which is a bit unnecessary, please reduce to 1 or no decimal places.

- corrected

7. Figure 3 – gene ACVR2A appears to be offset versus the others, can this be corrected.

- corrected

8. Line42-45 – This sentence is not clear and could be split into two, and also made the numerical data on line45 (2d) refers to microsatellite length.

- Sentence has been split and altered to clarify the message. Text has been added before numerical data to indicate that this refers to the microsatellite length on the mutation locus.

---

## [Decision Letter · Decision Letter 1]

10 Feb 2021

PONE-D-20-19308R1

Framing the potential of public frameshift peptides as immunotherapy targets in colon cancer

PLOS ONE

Dear Dr. Bins,

Thank you for submitting your manuscript to PLOS ONE. The reviewers were satisfied with the revision. However, reviewer 1 has requested several minor revisions to the text to clarify certain aspects of their work. Therefore, we invite you to submit a revised version of the manuscript that addresses them.

We look forward to receiving your revised manuscript.

Kind regards,

Jianjiong Gao

Academic Editor

PLOS ONE

Reviewers' comments:

Reviewer's Responses to Questions

**Comments to the Author**

1. If the authors have adequately addressed your comments raised in a previous round of review and you feel that this manuscript is now acceptable for publication, you may indicate that here to bypass the “Comments to the Author” section, enter your conflict of interest statement in the “Confidential to Editor” section, and submit your "Accept" recommendation.

Reviewer #1: (No Response)

Reviewer #2: All comments have been addressed

2. Is the manuscript technically sound, and do the data support the conclusions?

Reviewer #1: Yes

Reviewer #2: Yes

3. Has the statistical analysis been performed appropriately and rigorously? 

Reviewer #1: Yes

Reviewer #2: Yes

4. Have the authors made all data underlying the findings in their manuscript fully available?

Reviewer #1: Yes

Reviewer #2: Yes

5. Is the manuscript presented in an intelligible fashion and written in standard English?

Reviewer #1: Yes

Reviewer #2: Yes

6. Review Comments to the Author

Reviewer #1: The authors have revised the manuscript and have largely adressed reviewer concerns. I would request some minor revisions to the text to clarify certain aspects of their work to the reader.

1- 2.Clarification of Zscore

"As the Zscore is calculated by determining the mRNA transcript level of an individual mutated transcript in an individual Patient which is then related to the mRNA levels in the same transcript in the rest of the patient group we feel that there is an adequate internal control that negates the influence of copy number alteration and gene expression variations."

They should consider discussing the caveats of this approach better in the text. Probably as they say this would work fine for the copy number alterations, as the patients are all MSI and from the same cancer types. But it might be less appropriate with respect to the gene expression variation between individuals, for instance if the patients are from distinct gene expression clusters/subtypes within the cancer (I do appreciate however that MSI is typically considered a single subtype within a given cancer type). Moreover, the results are compared between three cancer types, and the global tissue-specific gene expression variation is not accounted for.

2- 2.Could the authors comment on the outliers in the MSS group in Fig 1 (TMB ~15000-20000) – are these potentially misclassified as MSS by TCGA? A comment to explain this should be added to the methods.

" -This is a very interesting question, and we believe that indeed it would be something that might be unclear to many readers. These outliers most likely represent a specific group of MSS patients with a mutation in polη, that has been described in literature to lead to high TMB, but no increased indel formation. A new section regarding MSI/MSS, TMB and polη has been added to the Methods."

Note that they are probably referring to POLE (epsilon) not polη (eta) -- two very different things.

They may consider citing https://www.sciencedirect.com/science/article/pii/S009286741731142X or https://www.ncbi.nlm.nih.gov/pmc/articles/PMC5849393/

or any other paper that previously reported extreme mutation burden in polymerase epsilon mutated tumors.

In addition, instead of qualifying them as "most likely" bearing a mutation in POLE, they should check if these tumors in fact bear one of the causal mutations.

3- Line 92 "we discarded the 70 least frequently occurring SNORFS (table 1)." and

Line 150: "In order to assess the PFP presentation on a protein level we are currently collecting patients samples in the ATAPEMBRO trial, which we aim to screen for specific PFP T-cell immunoreactivity by IFN-gamma Elispot."

With respect to using those SNORFS as a potential vaccine - would it make sense to consider substituting some of the NMD-SENS predicted from the top 20 SNORFS, by other NMD-RES or with a higher stringent deltaVAF threshold, even though they are less frequent? Of course, the NMD prediction is not always accurate, but it is improved with the VAF ratio. For instance, take the HDGF gene: it has a log2(TPM) of 6.11 (very highly expressed) but the deltaVAF is 66% which is not much different from 50% and is slightly less than the mean (67.93%), so it may not really be an NMD-escaping gene. Please discuss this and/or other similar examples.

4- As a consideration for future work and/or something to mention in the Discussion, they may consider checking if these SNORFS have NMD-inducing features of NMD endogenous targets (~5-20% of genes), some of which are different from the PTC position rules: 5' upstream ORF, intron in 3' UTR, long 3' UTR etc.

5- There may be issues with clarity of descriptions of the VAFratio method. In methods, it would help if they could show the formula used for VAFratio calculation along with some examples (one NMD-sensitive and one NMD-resistant). It probably does not make much sense to show the z-score formula (which they finally don't seem to use) while not showing the VAFratio formula. From the current description, one could infer that the VAF-ratio is higher for NMD-sensitive transcripts than for NMD-resistant transcripts, but they state: "The transcript independent method is actually the VAF-ratio, where a lower VAF ratio indicates NMD", which seems to be suggesting the opposite. Please take care to clarify this.

Also, "the RNA VAF only relies on the single allele that has been translated," and in the revisions-paragraph: "The fact that DNA contains two alleles, whereas just one is copied to the RNA" may be misleading, since both alleles are likely transcribed (while one may be degraded and the other no).

~end~

Reviewer #2: The authors have addressed all the comments of this reviewer, and have added additional figures, explanations, and corrections of minor points.

7. PLOS authors have the option to publish the peer review history of their article (what does this mean?). If published, this will include your full peer review and any attached files.

Reviewer #1: No

Reviewer #2: No

---

## [Author Response · Author response to Decision Letter 1]

28 Apr 2021

Response to reviewer questions

Final remarks

Reviewer #2 had no more comments. All comments below are concerning reviewer 1.

1. Clarification of Zscore

I. They should consider discussing the caveats of this approach better in the text. Probably as they say this would work fine for the copy number alterations, as the patients are all MSI and from the same cancer types. But it might be less appropriate with respect to the gene expression variation between individuals, for instance if the patients are from distinct gene expression clusters/subtypes within the cancer (I do appreciate however that MSI is typically considered a single subtype within a given cancer type). Moreover, the results are compared between three cancer types, and the global tissue-specific gene expression variation is not accounted for.

- First off al we thank reviewer 1 for this question and have tried to clarify the use of the Zscore within the text, stating that as suggested by the reviewer, this method is only suitable if used within a single population with the same molecular cancer subtype, as we did in our analysis, to avoid problems with copy number variation and gene expression variation.

2. POLE MSS patients

I. Could the authors comment on the outliers in the MSS group in Fig 1 (TMB ~15000-20000) – are these potentially misclassified as MSS by TCGA? A comment to explain this should be added to the methods.We thank reviewer #1 for pointing out that our explanation of the Zscore and VAF-ratio concepts are insufficient in the manuscript. The transcript independent method is actually the VAF-ratio, where a lower VAF ratio indicates NMD. We have edited both the main text, the legend of figure 3 and the methods section to better reflect the above. As stated below the NMD classification has been replaced by NMDetective algoritm by Lindenboom et al. (PMID: 31659324) Considering the downregulation of NMD resistant mutation this is also a good questions, and we are thankfull that the reviewer made us realize that the explanation for this fact does not become very clear. Most likely this downregulation is due to the effect of the VAF-ratio method, that relies on the difference between the DNA and RNA VAF. The fact that DNA contains two alleles, whereas just one is copied to the RNA it becomes expected that even in the situation of NMD resistant genes the mean VAF-ratio is approximately 50%. We have added a clarification of this in the methods section.

- We agree that with reviewer 1 that merely hinting at POLE for these patients is not sufficient. Therefore we have looked in detail at the mutational profile of these patients and concluded that these are indeed POLE patients. We have stated the exact muations in the text.

3. SNORFS as vaccine

I. With respect to using those SNORFS as a potential vaccine - would it make sense to consider substituting some of the NMD-SENS predicted from the top 20 SNORFS, by other NMD-RES or with a higher stringent deltaVAF threshold, even though they are less frequent?

2

- We see reviewer 1 has a valid point regarding the selection of the the SNORFS, but we believe that selecting on higher deltaVAF thresholds only makes sense for vaccination in a personalized vaccination strategy. In our aim for a single vaccination for a single cancer subgroup we aim to select more frequently occurring mutations with a sufficient deltaVAF over less frequently occurring mutations with a higher deltaVAF. We have tried to clarify this in the discussion.

4. NMD-inducing features of SNORFS

I. As a consideration for future work and/or something to mention in the Discussion, they may consider checking if these SNORFS have NMD-inducing features of NMD endogenous targets (~5-20% of genes), some of which are different from the PTC position rules: 5' upstream ORF, intron in 3' UTR, long 3' UTR etc.

- We thank reviewer 1 for this suggestion. This seems to be a very interesting avenue of further research and we are considering implementing it in our new projects. Nonetheless we feel that it might further complicate our current manuscript and therefore decided not to include it.

5. VAFratio calculation

I. There may be issues with clarity of descriptions of the VAFratio method. In methods, it would help if they could show the formula used for VAFratio calculation along with some examples (one NMD-sensitive and one NMD-resistant). It probably does not make much sense to show the z-score formula (which they finally don't seem to use) while not showing the VAFratio formula. From the current description, one could infer that the VAF-ratio is higher for NMD-sensitive transcripts than for NMD-resistant transcripts, but they state: "The transcript independent method is actually the VAF-ratio, where a lower VAF ratio indicates NMD", which seems to be suggesting the opposite. Please take care to clarify this. Also, "the RNA VAF only relies on the single allele that has been translated," and in the revisions-paragraph: "The fact that DNA contains two alleles, whereas just one is copied to the RNA" may be misleading, since both alleles are likely transcribed (while one may be degraded and the other no).We thank reviewer 1 for this suggestion. This seems to be a very interesting avenue of further research and we are considering implementing it in our new projects. Nonetheless we feel that it might further complicate our current manuscript and therefore decided not to include it.

- Thanks to reviewer 1 we have realized that the VAFratio can be cause for some confusion when reading our manuscript. We have tried to clarify our use by two measures. First off all, as suggested by reviewer 1, we have added the VAFratio formula to the methods section of the manuscript, including examples of NMD escaping and non-escaping transcripts. Furthermore we have altered figure 3b in order to not create confusion about the signs of NMD in the NMD resistant transcripts, by normalizing the data to the NMD resistant mutations. The prior decrease in NMD resistant mutations is probably caused by many factors including the inaccuracy of the NMD prediction algorithm that is based on mRNA transcript level information instead of mutation based mRNA information and therefore is

prone to missing exon skipping for example. We thinks this approach clarifies are underlying message that NMD sensitive transcripts are degraded more often, but that there is a substantial portion of mutated transcripts that does not get degraded by NMD and therefore can yield neo-epitopes.

---

## [Editor Report · Decision Letter 2]

30 Apr 2021

Framing the potential of public frameshift peptides as immunotherapy targets in colon cancer

PONE-D-20-19308R2

Dear Dr. Bins,

We’re pleased to inform you that your manuscript has been judged scientifically suitable for publication and will be formally accepted for publication once it meets all outstanding technical requirements.

Kind regards,

Jianjiong Gao

Academic Editor

PLOS ONE

---

## [Editor Report · Acceptance letter]

28 May 2021

PONE-D-20-19308R2 

Framing the potential of public frameshift peptides as immunotherapy targets in colon cancer 

Dear Dr. Bins:

I'm pleased to inform you that your manuscript has been deemed suitable for publication in PLOS ONE. Congratulations! Your manuscript is now with our production department. 

Kind regards, 

on behalf of

Dr. Jianjiong Gao 

Academic Editor

PLOS ONE